# Tree-Ring Analysis Reveals Density-Dependent Vulnerability to Drought in Planted Mongolian Pines

**ShouJia Sun** [1,2], **Shuai Lei** [1,2], **HanSen Jia** [1,2], **Chunyou Li** [3], **JinSong Zhang** [1,2] and **Ping Meng** [1,2,*]

1   Key Laboratory of Tree Breeding and Cultivation of State Forestry Administration, Research Institute of Forestry, Chinese Academy of Forestry, Beijing 100091, China; sunshj@caf.ac.cn (S.S.); 15907257267@163.com (S.L.); 15201438271@163.com (H.J.); zhangjs@caf.ac.cn (J.Z.)
2   Collaborative Innovation Center of Sustainable Forestry in Southern China, Nanjing Forestry University, Nanjing 210037, China
3   College of Landscape and Travel, Agricultural University of Hebei, Baoding 071000, China; lchy0815@163.com
*   Correspondence: mengping@caf.ac.cn; Tel.: +86-010-6288-9632

**Abstract:** Population density influences tree responses to environmental stresses, such as drought and high temperature. Prolonged drought negatively affects the health of Mongolian pines in forests planted by the Three-North Shelter Forest Program in North China. To understand the relationship between stand density and drought-induced forest decline, and to generate information regarding the development of future management strategies, we analyzed the vulnerability to drought of planted Mongolian pines at three stand densities. A tree-ring width index for trees from each density was established from tree-ring data covering the period 1988–2018 and was compared for differences in radial growth. Resistance ($R_t$), recovery ($R_c$), resilience ($R_s$), and relative resilience ($RR_s$) in response to drought events were calculated from the smoothed basal area increment (BAI) curves. The high-density (HDT) group showed a consistently lower tree-ring width than the border trees (BT) and low-density (LDT) groups. The BAI curve of the HDT group started to decrease five years earlier than the LDT and BT groups. Pearson correlation analysis revealed that the radial growth of all of the groups was related to precipitation, relative humidity (RH), potential evapotranspiration ($ET_0$), and standardized precipitation evapotranspiration index (SPEI) in the previous October and the most recent July, indicating that Mongolian pine trees of different densities had similar growth–climate relationships. Over the three decades, the trees experienced three severe drought events, each causing reduced tree-ring width and BAI. All of the groups showed similar $R_c$ to each drought event, but the HDT group exhibited significantly lower $R_t$, $R_s$, and $RR_s$ than the BT group, suggesting that the HDT trees were more vulnerable to repeated drought stress. The $RR_s$ of the HDT group decreased progressively after each drought event and attained <0 after the third event. All of the groups showed similar trends regarding water consumption under varying weather conditions, but the HDT group showed significantly reduced whole-tree hydraulic capability compared with the other two groups. From these results, HDT trees exhibit ecophysiological memory effects from successive droughts, including sap flux dysfunction and higher competition index, which may prevent recovery of pre-drought growth rates. HDT trees may be at greater risk of mortality under future drought disturbance.

**Keywords:** *Pinus sylvestris* var. *mongolica*; decline; tree ring; radial growth; recovery; resilience

## 1. Introduction

Population density and climate are important regulators of forest growth and competition [1,2]. Tree population density, a metric of tree abundance in a given area, is used as an indirect measure of competition intensity, which influences tree growth, dieback, and mortality [3]. A report by the Intergovernmental Panel on Climate Change (IPCC) affirmed that, from 1880 to 2012, the global average temperature increased by 0.85 °C and the trend for global warming accelerated rapidly in recent decades, particularly at high altitudes [4]. Climate change is expected to increase the frequency and severity of extreme climatic events (e.g., severe droughts, extreme temperatures, and heat waves) [5], resulting in a dramatic loss of biomes that are sensitive to these changes [6,7]. Unsurprisingly, reports of drought-induced forest mortality increased over recent years [8–10]. With the trend of global aridification ongoing, an increasing number of forests, especially high-density stands, are expected to be affected by drought stress [11]. Frequent and intense stress on trees may exceed their growth thresholds, thus eliciting uncharacteristic responses and causing deterioration. Nevertheless, although trees frequently experience deleterious stresses (e.g., drought, high temperatures, or pests), individual trees are able to recover and regain their pre-impact conditions [12]. Increasing evidence suggests that drought events may produce significant ecophysiological memory effects [13]. However, to date, the specific responses of trees in stands of different densities to repeated drought events remains poorly understood [14,15].

Tree growth is an indicator of tree vitality and reactions to environmental stress [16]. The post-impact recovery of trees is a long process that may involve time-lag or legacy effects [17,18]. Tree rings carry continuous, high-resolution information on tree growth over consecutive years. Given that this information accurately reflects environmental changes during tree growth [19], tree rings are widely used in studies on the resilience of forests in response to droughts [3,9]. A tree that exhibits vulnerability to drought stress is likely to recover unsatisfactorily after drought. Prolonged drought may reduce the recovery ability of a tree as a result of hydraulic failure [20,21] or carbon starvation [22]. When the accumulated stress exceeds the threshold of a tree, it may never recover to its pre-stress condition, thereby causing decline and even death [23,24].

Mongolian pine (*Pinus sylvestris* var. *mongolica*) is native to the Hulunbuir Sandy Lands and the Great Khingan Mountains in North China. Given that Mongolian pine is well adapted to dry and cold conditions, it was selected as a major species for development of the Three-North Shelter Forest program (TNSF) in North China, which now covers an area of >300,000 ha. However, continued warming, aridification, and the occurrence of severe droughts in recent decades had negative impacts on these trees. Extensive decline in the condition of Mongolian pines is apparent in some shelterbelts that were planted as part of the TNSF. A survey of Liaoning Province revealed that 62.75% of the trees in the Mongolian pine forests (in total 38,300 ha, of which the majority were high-density stands) declined in condition [25]. Previous studies on this species focused on the relationship between growth and climatic variables [26], intrinsic water use efficiency [27], sap flow and canopy water use [28], hydraulic limits underlying the low growth rate [21], and stand decline resulting from land use [29]. However, few studies investigated the response and adaptation of Mongolian pine in stands of different densities to drought, especially severe drought events. This paucity of information has hindered predictions of the growth, decline, regeneration, and management of plantation forests in a changing climate.

Therefore, we conducted a tree-ring analysis on the relationship between drought stress, growth, and the resilience of Mongolian pine in forests of three densities in Zhangbei County (Hebei Province, China). Generally, the natural conditions in the study region are unfavorable for Mongolian pine compared with those of its native habitat (Hulunbuir). Some trees in the high-density (HDT) stand showed signs of decline, whereas others in the low-density (LDT) stand and border trees (BT) did not. We hypothesized that trees grown at different densities would differ in vulnerability (i.e., resistance, recovery, and resilience) to drought, and that the decline of the HDT group would be attributable to a cumulative stress effect. The objectives of this study were to (1) evaluate the growth differences of Mongolian pine at three stand densities, (2) determine differences in post-drought recovery of the

stands, and (3) explore the link between stress accumulation and decline development. The results are expected to aid in prediction of future growth trends for these trees and development of climate-adapted management strategies for Mongolian pine.

## 2. Materials and Methods

### 2.1. Experimental Site and Sample Collection

The study was conducted in Zhangbei County, Hebei, China (Figure 1), which is located at the southern margin of the Inner Mongolian Plateau and is characterized by a moderate-temperate continental monsoon climate. The sampled stands of Mongolian pine were located at Ertai Town Forest Farm (41.33° N, 114.87° E, elevation 1385 m). The soil at the site was predominantly sandy soil with zonal chestnut soil and subject to severe wind erosion. The soil pH was 6.40–7.67 and contained $7.56 \pm 1.14$ g·kg$^{-1}$ of organic matter, $0.38 \pm 0.04$ g·kg$^{-1}$ of total phosphorous, $0.94 \pm 0.16$ g·kg$^{-1}$ of total nitrogen, and $0.99 \pm 0.10$ g·kg$^{-1}$ of total potassium. Three densities of Mongolian pine growing at the experimental location were assessed, namely, high-density stands (HDT) with 3 m × 3 m intra-spacing, low-density stands (LDT) with 4 m × 5 m intra-spacing, and border trees (BT) growing on the edge of the stands or on the sides of roads with neighboring natural grassland. The border trees showed healthy growth and the LDT group showed slight competition (the crowns slightly overlapped), whereas the HDT group exhibited strong competition (needle thinning and shortening, death of some mid- and lower-canopy shoots, and extending toward the crown).

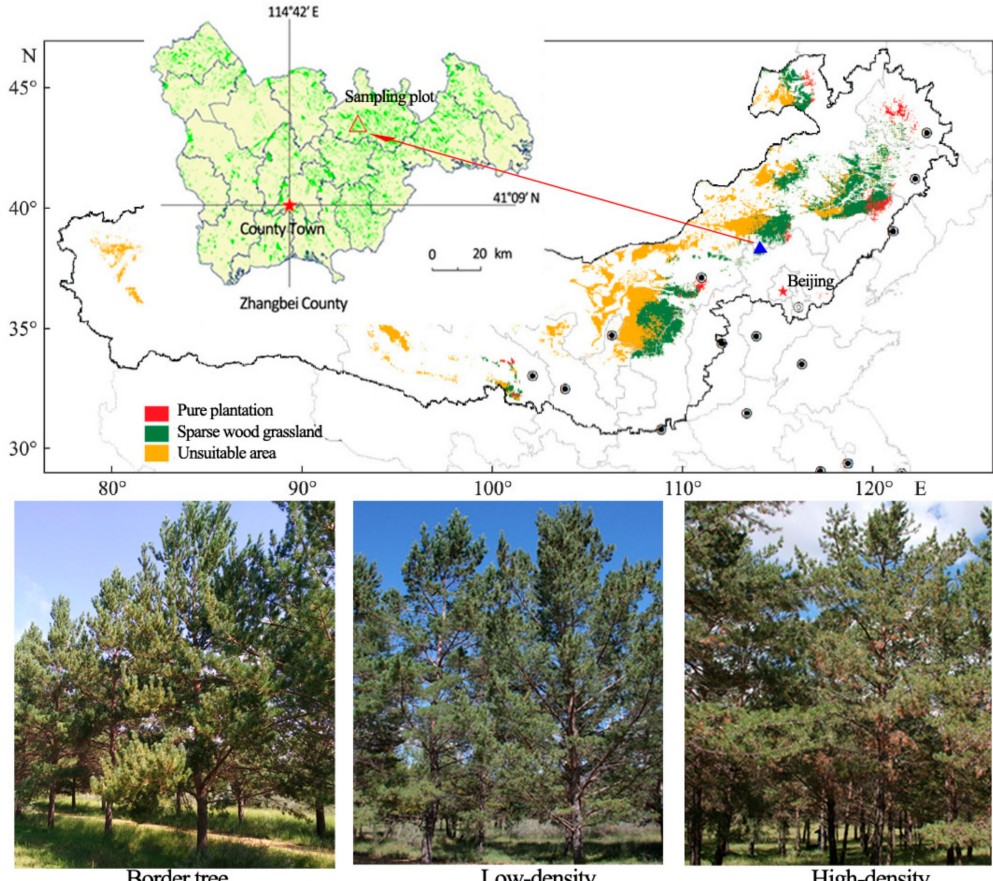

**Figure 1.** Classification of afforestation for Mongolian pine in sandy land in the construction regions of the Three-North Shelter Forest. Location, elevation, and climate of the experimental site in Zhangbei County, Hebei Province, China. Triangle: sampling plot for analysis of tree rings; red star: Zhangbei town.

## 2.2. Meteorological Data

Meteorological data (e.g., temperature, precipitation, relative humidity, and wind speed) for the period 1976–2018 were obtained from the county meteorological station (40.15° N, 114.70° E, elevation 1393.3 m). A Mann–Kendall test and the double mass curve method confirmed that the data were free of random variation and, thus, were representative of the local climatic trends.

Potential evapotranspiration ($ET_0$) was calculated using the Penman–Monteith Equation (1) [30]:

$$ET_0 = \frac{0.408\Delta(R_n - G) + \gamma\frac{900}{T+273}\mu_2(e_s - e_d)}{\Delta + \gamma(1 + 0.3\mu_2)}, \tag{1}$$

where $ET_0$ (in mm·d$^{-1}$) represents potential evapotranspiration, $R_n$ (in MJ·m$^{-2}$·d$^{-1}$) is the net radiation from plant surfaces, $\gamma$ (kPa·°C$^{-1}$) is the psychrometric constant, $\Delta$ is the slope of the saturated vapor pressure-temperature relationship, $T$ is the mean air temperature, $\mu_2$ is the wind speed at 2 m above the ground, $e_s$ is the saturation vapor pressure, and $e_d$ is the vapor pressure.

A standardized precipitation evapotranspiration index (SPEI) [31] was used to assess the degree of drought in the area:

$$\text{SPEI} = W - \frac{C_0 + C_1W + C_2W^2}{1 + d_1W + d_2W^2 + d_3W^3}, \tag{2}$$

where $W = \sqrt{-2\ln(P)}$ for $P \leq 0.5$, and $P$ is the probability of exceeding a determined $D$ value, $P = 1 - \text{F}(x)$. If $P > 0.5$, then $P$ is replaced by $1 - P$ and the sign of the resultant SPEI is reversed. The timescales assessed were 1 month and 12 months. An annual 12-month SPEI of <0 for more than three continuous years was considered a severe drought event. Using this criterion, three severe drought events were identified: 1997–2001, 2006–2009, and 2011–2014.

## 2.3. Competition Index, Tree-Ring Width, and Chronology

Four plots (20 m × 20 m) were established in each of the three mature Mongolian pine plantation stands. Plantations minimized between-tree differences within the same density plot (e.g., same ages and provenances, and similar heights and sizes). For each plot, all trees were numbered, then the diameter at breast height (DBH) and height were measured. Finally, a total of 12 sample plots in the three stands were surveyed. The stand density, mean height, and mean DBH were calculated from the number of trees (Table 1). Competition indices (CI) for each subject tree were calculated using Hegyi's Equation (3) [32]:

$$\text{CI} = \sum_{j=1}^{n}\left(\frac{D_j}{D_i} \cdot \frac{1}{d_{ij}}\right), \tag{3}$$

where $D_j$ is the DBH (cm) of neighboring trees in a circle with a radius of 8 m from the subject tree, $D_i$ is the DBH (cm) of the subject tree, and $d_{ij}$ is the horizontal distance (m) between neighboring trees and the subject tree.

**Table 1.** Statistical description (mean ± SD) of border trees, low-density, and high-density stands of Mongolian pine at the experimental site in Zhangbei County, Hebei Province, China. Lower-case letters indicate significant differences ($p < 0.05$).

|  | Mean Diameter at Breast Height (DBH) (cm) | Mean Height (m) | Stand Density (Stems/ha) | Hegyi's CI | No. of Cores |
|---|---|---|---|---|---|
| BT | 25.23 ± 2.87 a | 11.3 ± 1.7 a | 250 ± 7 c | 0.53 ± 0.13 c | 24 (48) |
| LDT | 22.51 ± 2.25 b | 10.8 ± 1.4 a | 487 ± 39 b | 2.03 ± 1.12 b | 24 (48) |
| HDT | 15.91 ± 2.14 c | 8.7 ± 1.1 b | 985 ± 82 a | 4.26 ± 0.78 a | 24 (48) |

In each of the three stand densities, 6 trees in each plot and a total of 24 trees were randomly selected. Two core samples were collected with an increment borer from each tree at breast height (1.3 m above ground) and immediately sealed in containers. The samples were fixed and dried in the laboratory, polished with sandpaper to the desired surface finish, and cross-dated visually with skeleton plots. Tree-ring width (TW) (Supplementary Materials File S1) was measured with a LINTAB™ 6 measuring system (RINNTECH, Heidelberg, Germany) to 0.01 mm precision, cross-dated with COFECHA software, and verified to eliminate potential errors [33]. Long-term trends caused by aging were removed mostly using regional curve standardization (RCS) with ARSTAN 4.4 software [34] to develop a residual chronology (RES). The basal area increment (BAI) was calculated using Equation (4) [35]:

$$\text{BAI} = \pi\left(R_n^2 - R_{n-1}^2\right), \tag{4}$$

where *R* is the tree radius and *n* is the year of tree-ring formation.

### 2.4. Sap Flow Velocity

From 2015 to 2017, three sample trees were selected from each of the groups and thermal dissipation probes were inserted into the south and north sides of the tree trunk at breast height (1.3 m). Three dataloggers (CR10X, Campbell Scientific, Logan, UT, USA) were used to record the data of eighteen thermal probes in total, because the three stands were not together. Sap flow velocity was calculated using the empirical Equation (5) [36]:

$$Fs = 0.0119\left[\frac{D_{TM} - D_T}{D_T}\right]^{1.231} \times As \times 3.6, \tag{5}$$

where *Fs* (in L·h$^{-1}$) is the sap flow velocity, *As* (in cm$^2$) is the sapwood area, $D_{TM}$ is the maximum temperature difference within a day, and $D_T$ is the temperature difference at a given moment.

### 2.5. Vulnerability to Drought

The response of the Mongolian pine trees to drought was quantified by means of a BAI-based method [37]. We assumed that the negative effect of antecedent drought events on vigor was lasting and, thus, would exhibit a legacy effect. The following four indices describing the response of the trees to drought were calculated following Lloret et al. [38].

Resistance ($R_t$) is the ratio between BAIs during and before drought, representing the ability of a tree to maintain growth under drought stress. Recovery ($R_c$) is the ratio between BAIs after and during drought, describing the ability to regrow after drought. Resilience ($R_s$) is the ratio between BAIs after and before drought, representing the ability to regain the pre-drought growth rate. Relative resilience ($RR_s$) is defined as $R_s$ weighted by the damage incurred during the drought. The indices were estimated as follows:

$$\text{Resistance } R_t = \text{BAI}_D/\text{BAI}_{\text{pre}}, \tag{6}$$

$$\text{Recovery } R_c = \text{BAI}_{\text{post}}/\text{BAI}_D, \tag{7}$$

$$\text{Resilience } R_s = \text{BAI}_{\text{post}}/\text{BAI}_{\text{pre}}, \tag{8}$$

$$RR_s = (\text{BAI}_{\text{post}} - \text{BAI}_D)/\text{BAI}_{\text{pre}}, \tag{9}$$

where $\text{BAI}_D$ is the average BAI during a drought event and $\text{BAI}_{\text{pre}}$ and $\text{BAI}_{\text{post}}$ are the average BAIs during the three years before and after the drought event, respectively. Given that drought events and impacts often span multiple years, $\text{BAI}_D$ included multiple years where appropriate [37].

### 2.6. Data Analysis

Tree growth each year is affected not only by the meteorological factors for that year, but also those of previous years. Therefore, climate response was analyzed for each year using data covering

the period from June of the previous year to October of the year in question. The growth–climate relationships were investigated using the treeclim R package [39]. Tree-ring width, BAI, sap flux, and vulnerability among the three density groups were analyzed using ANOVA and the least significant difference (LSD) test with IBM SPSS Statistics 20 software (IBM Corporation, Armonk, NY, USA). Probabilities were considered statistically significant at $p < 0.05$.

## 3. Results

### 3.1. Changes in Meteorological Factors and $ET_0$

Between 1988 and 2018 (Figure 2), the total annual precipitation was <533.9 mm (mean: 388.5 mm) and fluctuated considerably, with wet and dry years recorded periodically. The minimum precipitation occurred in 1997 (245.2 mm). Overall, total annual precipitation tended to decrease, with a reduction of 56.7 mm over the 30-year period. The mean annual temperature varied from 3.1 to 5.3 °C and tended to rise significantly ($R^2 = 0.23$, $p < 0.01$), with a total increase of 0.95 °C over the 30-year period. The mean annual relative humidity fluctuated from 52% to 61% and showed a weakly increasing trend. The $ET_0$ exhibited a weakly decreasing trend rather than increasing with warming, possibly attributable to changes in the local vegetation cover as a result of TNSF tree decline or death. Over the 30-year period, the mean annual SPEI fluctuated from −1.13 to 1.52 and showed a strong decreasing trend. The SPEI trend indicated that the region became increasingly dry over the previous 30 years. Therefore, the meteorological data indicated a trend of local aridification over the period, as evidenced by a reduction in precipitation and a rise in temperature.

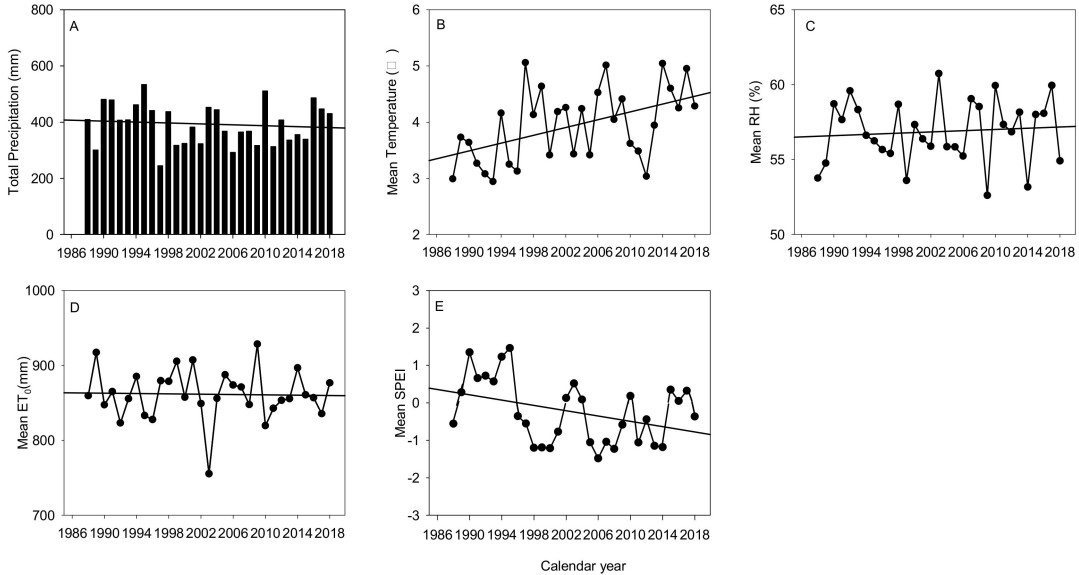

**Figure 2.** Variation in total precipitation (**A**), mean temperature (**B**), mean relative humidity (**C**), mean potential evapotranspiration ($ET_0$) (**D**), and mean standardized precipitation evapotranspiration index (SPEI) (**E**) during the study period at Zhangbei County, Hebei Province, China, at an annual timescale.

### 3.2. Tree-Ring Width

The HDT (15.91 ± 2.14 cm) group showed a smaller DBH than the LDT (22.51 ± 2.25 cm) and BT (25.23 ± 2.87 cm) groups, and all of the groups showed significant differences in DBH. The tree height of the HDT (8.7 ± 1.1 m) group was shorter than those of the LDT (9.8 ± 1.4 m) and BT (11.3 ± 1.7 m) groups, but the difference was only statistically significant between the HDT and BT groups. The competition index of the HDT (4.26 ± 0.78) was the highest, followed by the LDT (2.03 ± 1.12) and BT (0.53 ± 0.13) groups. The differences were significant among the three groups (Table 1).

The tree-ring widths of the three groups (Figure 3A) exhibited an initially rising trend with a subsequent fluctuating decrease. All peaks were recorded in 1993. Over the study period, the HDT group showed a consistently smaller tree-ring width than the other groups. However, after 2009, the HDT group showed a lower tree-ring width index than the LDT and BT groups. The 3-year moving average of the tree-ring width data revealed that the three groups of trees experienced three low-growth phases. The 1988–2018 period was thus divided into four stages (Figure 3B) in accordance with the endpoint of each drought event (i.e., 2001, 2009, and 2014). In Stages I and II, the LDT group showed no significant differences in tree-ring width compared with the other two groups, whereas the differences between the HDT group and the BT group were statistically significant ($p < 0.05$). However, in Stages III and IV, the HDT group showed highly significant lower tree-ring widths than the other two groups ($p < 0.05$; Figure 3C). The residual (RES) tree-ring indexes of all of the groups followed similar trends of variation, indicating that they responded in similar ways to climate change (Figure 3D).

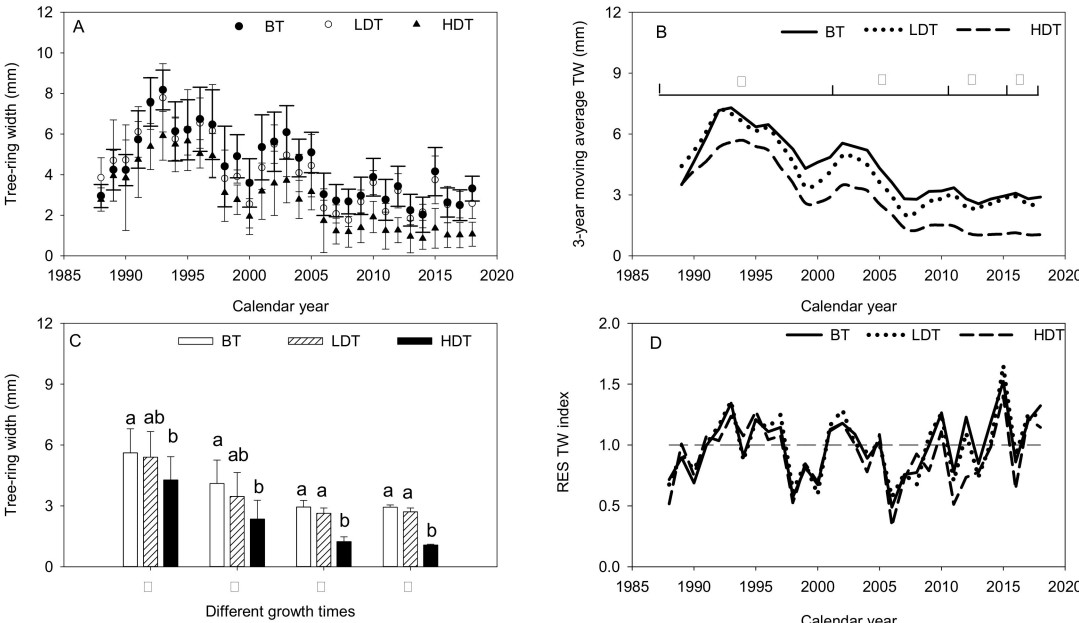

**Figure 3.** Differences in tree-ring width (**A**) and 3-year moving average tree-ring width (**B**) among three densities of Mongolian pine trees. Difference in tree-ring widths among the three density groups in different growth periods (**C**). Lower-case letters indicate significant differences at $p < 0.05$. Residual tree-ring index among the three densities of the stands (**D**).

*3.3. BAI*

For all of the groups, the BAI followed a highly significant quadratic relationship with age ($p < 0.01$) (Figure 4A); the HDT group peaked in 2004, whereas the other two groups peaked in 2009. The BAI data revealed that three drought events occurred during the period, creating three valleys (Figure 4B). After the end of each drought event, the BAIs of the LDT and BT groups recovered to or exceeded the pre-drought level. In contrast, the BAI of the HDT group exceeded the pre-drought level after the first drought event but was lower than that level after the second and third events. In Stage I, the HDT group showed a significant difference in BAI with the other two groups, but the difference between the LDT group and the BT group was not significant ($p > 0.05$). However, in Stages II, III, and IV (Figure 4C), significant differences in BAI were observed among the three groups (all $p < 0.05$).

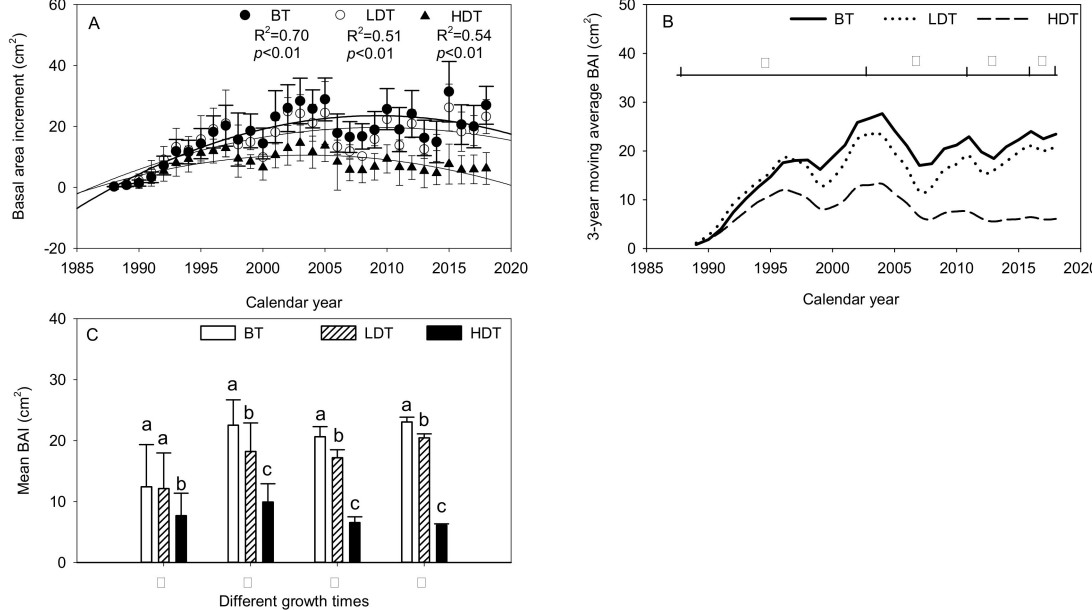

**Figure 4.** Differences in basal area increment (BAI) (**A**) and 3-year moving average BAI (**B**) between Mongolian pine trees of the three density groups. Difference in mean BAI (**C**) among the three densities in different growth periods. Lower-case letters indicate significant differences at $p < 0.05$.

### 3.4. Relationships among Tree-Ring Records and Climatic Variables

For the three groups of trees, the 3-year moving average tree-ring width (TW) was significantly correlated with temperature ($p < 0.05$) and showed a highly significant relationship with 12-month SPEI at an annual timescale ($p < 0.01$; Figure 5). The 3-year moving average BAI showed a significant relationship only with temperature ($p < 0.05$).

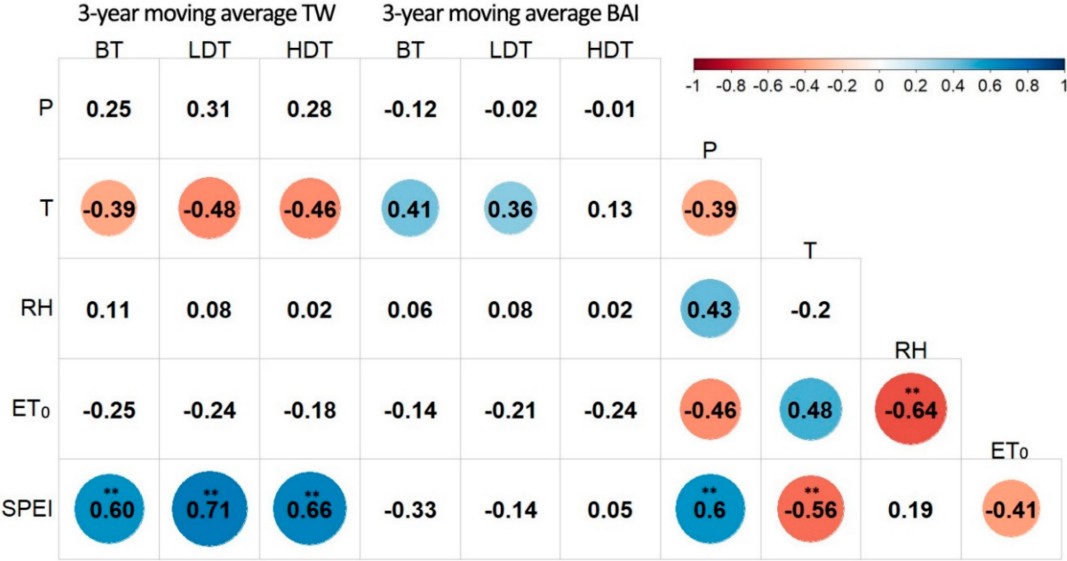

**Figure 5.** Pearson correlation coefficients for 3-year moving average tree-ring width (TW) and basal area increment (BAI) with mean annual climatic factors for Mongolian pine trees of the three density groups at an annual timescale.

Analyses of the relationships between growth and monthly climatic variables enabled assessment of the sensitivities of different stand densities to climatic constraints. For all of the groups, the RES tree-ring index showed negative associations with temperature in October of both the previous and

current years ($p < 0.05$). In particular, for the BT group, the association was highly significant for October of the previous year ($p < 0.01$). For all of the groups, the RES tree-ring index showed a significant positive association with relative humidity (RH), but showed a significant negative relationship with $ET_0$ in July of both the previous and current years ($p < 0.05$). For the HDT and LDT groups, the RES tree-ring index showed a significant positive association with precipitation in the most recent July and the previous October. For the BT group, the association was statistically significant in July of both the previous and current years ($p < 0.05$). For all groups, the RES tree-ring index showed a significant positive association with 1-month SPEI in the most recent July and in October of both the previous and current years (Figure 6). These results indicated that Mongolian pine trees of different densities had similar growth–climate relationships.

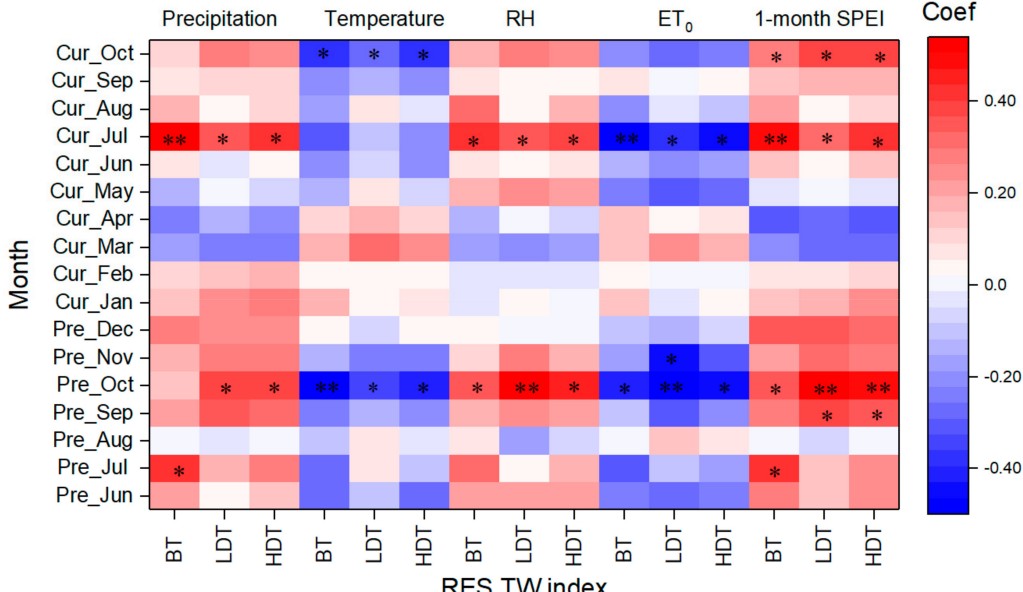

**Figure 6.** Relationships between the residual (RES) tree-ring index and monthly climatic factors for all of the groups at a monthly timescale. Cur_month represents the current year of ring formation and Pre_month represents the climate variables during the year preceding ring formation. Asterisks * and double asterisks ** indicate significant differences at $p < 0.05$ and $p < 0.01$, respectively.

### 3.5. Sap Flow

The sap flow velocities of the three groups (Figure 7A) exhibited similar periodical variations in August, starting at ~07:00 and peaking at ~12:00–15:00. However, the three groups differed substantially in their peak velocities (BT group: 32.04, 28.69, 19.96, 32.26, and 30.39 cm·h$^{-1}$; LDT group: 29.28, 27.38, 17.73, 29.95, and 27.67 cm·h$^{-1}$; HDT group: 15.98, 16.07, 6.62, 15.56, and 16.07 cm·h$^{-1}$), with average peak velocities of 53.50% and 49.05% recorded in the HDT group relative to the other two groups, respectively. The BT and LDT groups consumed 10.55–28.54 L and 9.88–25.98 L of water daily per tree, respectively, compared with 7.04–11.58 L consumed daily by the HDT group. All of the groups showed similar trends in water consumption under varying weather conditions, but the daily water consumption per tree by the HDT group was ~50.73% and ~46.31% of the LDT and BT groups, respectively. When compared at monthly intervals (Figure 7C), the HDT group consumed significantly less water than the other two groups. From April to October, the HDT group consumed only 55.03% and 52.83% of the water used by the other two groups (2838.13 ± 175.11 L vs. 1561.68 ± 83.69 L and 2956.27 ± 86.11 L, respectively).

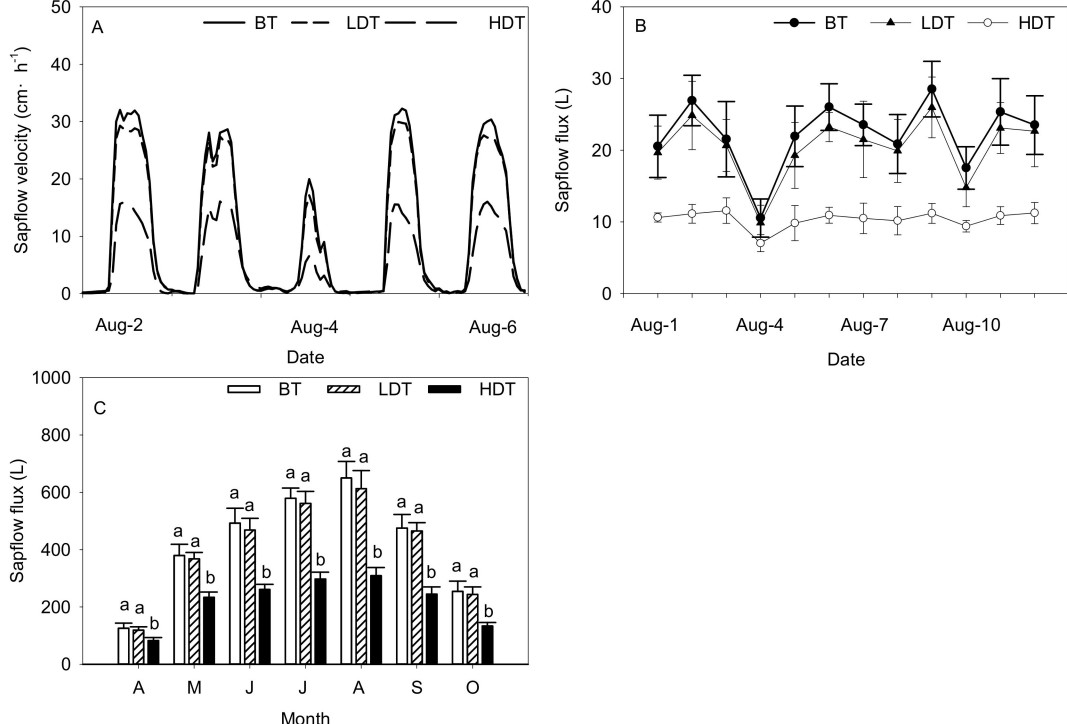

**Figure 7.** Sap flow velocity (**A**) and sap flow flux (**B**) among the three density groups of Mongolian pine trees. Sap flow flux among the BT, LDT, and HDT groups at a monthly timescale (**C**). Lower-case letters indicate significant differences at $p < 0.05$.

### 3.6. Resistance, Recovery, and Resilience Indices

The BAI data were analyzed for $R_t$, $R_c$, and $RR_s$ during three drought events. The HDT group showed no significant differences in $R_c$ values with the LDT and BT groups (Table 2), suggesting that $R_c$ may be associated with the inherent characteristics of the tree species. The $R_t$ values of the HDT group were significantly reduced (by 34.20% for 1997–2000, 20.87% for 2006–2008, and 17.24% for 2011–2013) compared with those of the BT group. The $R_s$ values of the HDT group were significantly reduced (by 40.18% for 1997–2000, 37.73% for 2006–2008, and 35.90% for 2011–2013) compared with those of the BT trees. The $RR_s$ values of the HDT group (0.35 for 1997–2000, 0.01 for 2006–2008, and −0.01 for 2011–2013) were significantly lower than those of the other two groups during the three drought events.

**Table 2.** Resistance ($R_t$), recovery ($R_c$), resilience ($R_s$), and relative resilience ($RR_s$) among the three density groups of Mongolian pine trees. Lower-case letters indicate significant differences at $p < 0.05$.

| Drought Event Year | $R_t$ | | | $R_c$ | | | $R_s$ | | | $RR_s$ | | |
|---|---|---|---|---|---|---|---|---|---|---|---|---|
| | BT | LDT | HDT | BT | LDT | HDT | BT | LDT | HDT | BT | LDT | HDT |
| 1997–2000 | 1.05 ± 0.27a | 0.84 ± 0.19b | 0.79 ± 0.14b | 1.51 ± 0.23a | 1.58 ± 0.24a | 1.45 ± 0.15a | 1.59 ± 0.21a | 1.32 ± 0.13b | 1.14 ± 0.11b | 0.54 ± 0.12a | 0.48 ± 0.14a | 0.35 ± 0.10b |
| 2006–2008 | 0.71 ± 0.17a | 0.62 ± 0.21b | 0.58 ± 0.18b | 1.16 ± 0.27a | 1.28 ± 0.16a | 1.02 ± 0.20a | 0.82 ± 0.27a | 0.79 ± 0.18a | 0.60 ± 0.14b | 0.12 ± 0.04a | 0.17 ± 0.06a | 0.01 ± 0.06b |
| 20112013 | 0.92 ± 0.14a | 0.95 ± 0.11a | 0.78 ± 0.09b | 1.16 ± 0.23a | 1.23 ± 0.22a | 0.99 ± 0.19a | 1.07 ± 0.16a | 1.16 ± 0.21a | 0.78 ± 0.14b | 0.15 ± 0.07a | 0.21 ± 0.09a | −0.01 ± 0.05b |

## 4. Discussion

### 4.1. Radial Growth

Forests are among the largest biomes on Earth. Only healthy forests provide beneficial ecological services and play a positive role in carbon storage, biodiversity conservation, and climate regulation [40,41]. High population density increases competition between trees for limited resources.

Climate warming and continued drought may exacerbate drought stress in water-deficient regions and therefore negatively affect tree growth. Drought stress causes physiological harm to trees [20]. When physiological harm exceeds the threshold for some trees, irreversible damage occurs, such as defoliation and dieback [42]. In response to deleterious environmental factors (e.g., drought and high temperatures), trees may reduce their vigor or growth rate, undergoing decline and die-off [43,44].

It is important to investigate the effects of competition and climate on forest growth, as these are two of the main forces affecting forest dynamics. In the same region, the effect of competition is greater than that of climate [2]. Forest growth decline is exacerbated by high density, particularly during periods of severe drought [45]. In the present study, compared with those of the LDT and BT groups, the HDT group showed significantly lower (by 41.48% and 58.39%, respectively) DBH and smaller TW. A previous study observed that poplar (*Populus deltoides* × *Populus. nigra*) trees grown at a high density displayed a lower stem circumference [46]. Continued drought similarly slowed the radial growth of Scots pine [47] and induced a significantly lower growth rate in *Quercus cerris* and *Quercus. pubescens* trees that suffered decline (in comparison with non-declining trees) [23]. Even in mixed forests, growth of both species was reduced due to competition overriding any complementary advantage when intraspecific competition increased [48]. A negative trend in BAI is a strong indicator of slowing tree growth [35]. In the present study, the HDT group exhibited a significantly lower BAI (Figure 4C) and a higher Hegyi's CI than the LDT and BT groups, which were mainly associated with density-related competition. Less growth was considered to be an early-warning signal for future drought-induced death. Song et al. [49] reported that damaged trees in Mongolian pine plantation forests exhibited an abrupt decrease in BAI between 1990 and 1996, whereas their counterparts in natural forests did not show this phenomenon. The authors observed that a decrease in groundwater level subjected the plantation forests to severe water stress, causing their deterioration. Plants commonly respond to variation in resource availability by gaining or losing biomass, [42], however, in this research, biomass loss failed to prevent tree decline.

During the three severe drought events, all of the groups experienced substantial reductions in growth rate. It was widely reported that drought, high temperature, and climatic extremes retard plant growth [5,11]. Nardini et al. [50] studied the behavior of woody species during an extremely dry summer and observed a negative impact of the drought on water relationships and non-structural carbohydrates. The extreme summer drought of 2005 reduced plant biomass production, increased plant mortality, and slowed plant growth in Amazonian rainforests [51]. The extreme 2003 summer drought that affected large areas of Europe almost immediately impaired plant growth [16]. Suarez et al. [52] reported that the severe drought of 1998–1999 in northern Patagonia impaired tree growth; adult trees with declining growth were particularly vulnerable to drought, therefore, mean growth rate was a good predictor of mortality in adult trees. In the present study, the HDT group was indicated to be more severely affected by the drought events than was the LDT and BT groups. When drought stress exceeded the recovery of some Mongolian pine individuals, the trees exhibited dieback, defoliation, and decelerated growth. These drought-induced stand changes are known to disrupt the composition and ecological services of forests [53].

Environmental variables impair tree growth [54], of which precipitation, temperature, and SPEI are critical factors. Precipitation alleviates drought stress and supports plant growth. In the current study, for all groups, radical growth showed a significant positive association with precipitation in the growing season, with the relationships being statistically significant in the most recent July. This result was consistent with previous findings on the relationship between precipitation and growth of natural Mongolian pine [49]. Rubio-Cuadrado et al. [55] similarly observed a positive association between the TW of Scots pine and precipitation in May. All groups exhibited negative associations between radical growth and temperature, with the relationships being statistically significant in October of both the previous and current years. Bao et al. [26] investigated natural Mongolian pine in Hulunbuir and observed a negative association between growth rate and maximum temperature during April–September, which was likely attributable to the growth phenology of this species. A significant

negative association between the temperature in May and radical growth of Scots pine was observed, which also probably reflected the preference of this conifer for cold climates [55]. A strong, positive relationship was observed between SPEI and radical growth for all of the groups in the current July and in October of both the previous and current years, indicating that the growth of Mongolian pine was severely limited by water deficit. The significant SPEI–growth associations indicated short-term responses of the plant at arid sites, revealing species-specific growth responses to drought [56].

Climate–growth relationships of Mongolian pines showed that the meteorological factors in the previous October and the current July had significant impacts on the radial growth. Some previous studies observed that pine growth began before the needles unfolded, and nonstructural carbohydrates (NSC) storage pools in late winter correlated positively with spring growth [57,58]. Earlywood formation of Mongolian pine trees probably used young reserves that accumulated in the previous October. In the current July, precipitation, relative humidity, and SPEI increased, which probably promoted the formation of latewood. Oberhuber et al. [59] also found that starch reached a maximum in mid-July, which contributed to late wood formation of Scots pine. However, all groups showed similar sensitivities to climatic factors at the same site. A previous study showed that the effect of competition was greater than that of climate in the same area [2], indicating that density is an important factor for the growth of Mongolian pine at a given site.

### 4.2. Vulnerability to Drought

Recently, global tree deterioration and death affected the health and ecological services of forests [41,60]. Decline and increasing mortality of forests may be associated with thresholds in specific components that affect the trees' ability to regain pre-stress growth performance. Understanding such abilities in forests of different densities is critical for forest management. Under drought stress, some trees adapt via internal regulatory mechanisms, with growth recovering once the stress disappears [5,61]. Moreover, repeated mild drought events increase resistance toward extreme drought stress [62]. In this study, all of the groups showed similar $R_c$ values (Table 2) but significantly different $R_t$ values. During each drought event, the HDT group showed lower $R_t$ than the BT group, indicating that the growth of the HDT group was more seriously affected by drought. Galiano et al. [13] suggested that, if both the $R_t$ and $R_c$ of a tree were partly dependent on stored reserves, the two indices would suffer a tradeoff after drought. According to this reasoning, the lower observed $R_t$ in the HDT group may be explained by the competition for limited water and nutrients. However, a previous study illustrated that young trees showed higher $R_t$, but not necessarily greater $R_s$, compared with adult trees [55]. Moreover, naturally growing pines were reported to show greater $R_t$ to drought than planted trees. These results indicated that the response of trees to changes in environmental conditions was complex [38].

Resilience characterizes the ability of a tree to regain its original growth rate after interference from an extreme event [12]. A value of $R_s < 1$ reflects the continuing effect of that event. In the present study, after the second and third drought events, the HDT group exhibited a significantly reduced $R_s$ value compared with the other two groups, which was consistently $<1$, indicating that each event negatively impacted the HDT group, thereby preventing the trees from regaining their pre-drought growth rates.

Relative resilience is defined as $R_s$ weighted by the damage incurred during drought. If the growth rate after the event was lower than that during the event, $RR_s$ was $<0$. In the current study, after each drought event, the HDT group showed a significantly lower $RR_s$ value compared with the HDT and BT groups, which decreased stepwise and attained $<0$ after the third drought event. This finding indicated that, over the 30-year period, the HDT group experienced accumulated stress, thereby causing a loss of vigor and growth collapse (i.e., the post-drought growth rate was less than the growth rate during the drought) [63]. Taeger et al. [64] reported that Scots pine of different provenances differed substantially in $R_s$, and trees of different ages showed similar $R_s$ values that did not decrease with age. Therefore, increased forest mortality may be associated with thresholds in specific components of tolerance ($R_t$, $R_c$, and $R_s$) rather than an overall deterioration of drought response [38]. Supporting this argument, in

the present study after each drought event (Table 2), the HDT group largely retained its $R_s$. After the third drought event, the $RR_s$ of the HDT group decreased to <0, indicating that the trees were affected by the accumulation of drought stress.

A previous study reported that the crown diameter of Mongolian pine increased with both age and stand density [65]. As crowns may overlap, competition is a crucial factor that affects Mongolian pine growth [66]; thus, trees growing in a dense forest may undergo decline more readily. Trees in low-density stands were reported to show greater $R_t$ and $R_c$, indicating that trees growing at lower densities were less vulnerable to the harmful effects of drought [3] and high-density stands were more sensitive to climatic fluctuations (particularly precipitation). Plant responses to an extreme event may involve a legacy effect [17]. Some years after the beginning of a drought event, the response of HDT trees to the event may become more noticeable, which would partly explain why repeated stress events impair their ability to recover.

Decline of Mongolian pine trees is closely associated with water status [29]. Sap flow is an important indicator of tree health [67]; it is essential to maintain the transpiration stream and was recognized as crucial to understand the hydraulic function (and dysfunction in particular environments) of trees [68]. A previous study of Norway spruce observed that, compared with healthy trees, unhealthy trees maintained only one-third of the sap flow. Sap flow density in the low-density stands (thinned) was significantly higher than the non-thinning stands [69]. In the current study, the sap flow velocity of the HDT group was ~50% lower than the LDT and BT groups, revealing the poor hydraulic conductance of the HDT group and resulting in these trees growing more slowly [70]. The differences in sap flow between healthy and unhealthy trees revealed that ongoing multi-year forest decline was mainly driven by loss of whole-tree hydraulic capability, which in turn limited assimilation capacity [71]. Subsequently, the HDT group may have become more vulnerable to drought, experiencing greater hydraulic failure [21] and carbon starvation [22]. Consequently, the HDT trees faced a higher risk of death in the future.

## 5. Conclusions

In the TNSF stands, high density and drought adversely affected the growth of mature Mongolian pine trees, resulting in lower TW and BAI values and earlier onset of BAI decrease in the HDT group compared with the LDT and BT groups. All of the groups showed similar $R_c$ values during the three severe drought events, but the HDT group exhibited distinctly lower $R_t$, $R_s$, and $RR_s$ values than the LDT and BT groups. After each drought event, the HDT group exhibited a slowing of radial growth, indicating the vulnerability of this group to drought stress. The HDT group showed progressively lower $RR_s$, eventually attaining <0 at the third drought event, suggesting that the accumulation of the impact of drought stress caused their decline (i.e., their inability to regain the pre-drought growth rate after the drought). Sap flow data also revealed hydraulic dysfunction in the HDT trees. These results suggest that, due to the cumulative effects of drought stress and a higher competition index, HDT trees face an increased risk of future mortality.

**Supplementary Materials:** The following are available online at http://www.mdpi.com/1999-4907/11/1/98/s1, File 1: Tree-ring data.

**Author Contributions:** S.S., J.Z., and P.M. conceived the idea and contributed to the study design, discussed the results, and wrote the manuscript. S.L., H.J., and S.S. performed data collection in the field and contributed to chronology data analysis. C.L. and S.S. performed meteorological data collection and analysis. P.M. and J.Z. funded the study. All authors contributed to interpretation of the results, discussion, and approved the final manuscript. All authors have read and agreed to the published version of the manuscript.

**Funding:** This work was supported by a National Nonprofit Institute Research Grant of the Chinese Academy of Forestry (CAFYBB2018ZA001), the National Natural Science Foundation of China (grant no. 31470705), and the Project of the Co-Innovation Center for Sustainable Forestry in Southern China of Nanjing Forestry University.

**Acknowledgments:** The China Meteorological Data Service Center provided meteorological data. We thank Leonie Seabrook, and Robert McKenzie, from Liwen Bianji, Edanz Group China (www.liwenbianji.cn/ac), for editing the English text of a draft and polishing the revised manuscript.

**Conflicts of Interest:** The authors declare that the research was conducted in the absence of any commercial or financial relationships that could be construed as a potential conflict of interest.

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
