# Peer review of "Tree-Ring Analysis Reveals Density-Dependent Vulnerability to Drought in Planted Mongolian Pines"

_forests, doi:10.3390/f11010098_

Round 1

Reviewer 1 Report

The authors of Sun et al. resubmitted manuscript ‘Tree-ring analysis reveals density-dependent vulnerability to drought in planted Mongolian pines’ have advanced the work and clarified many of the points I, and the other reviewer, raised on the previous iteration of the work. Thanks for that effort. I also think that Figure S2 is a very good addition to the paper and should maybe be considered to be included in the main text, as I agree with Reviewer #2 that mixing tree ring width, BAI, and tree ring indices (and actually not making clear when each one is used) is not a very good idea. In that sense, I like that Fig. S2 shows the higher decrease in growth in high density trees after controlling for the many unaccounted potential confounders, that’s very positive and would be worth discussing and adding to current Figure 3 as an extra panel. Despite the author’s effort, I don’t think this version of the manuscript is ready, my main concerns are:

The differences in the responses to ring width, ring indices and BAI are not very reassuring. Figure 5 shows opposed responses by width and BAI to temperature, which seems rather odd.0 Similarly, the contrast in responses between Figure 6, Figure S4, and the similarly-looking figure in the cover letter is strange, and should be discussed properly. How much of the temperature signal is long-term trends versus high frequency change is important to discuss. The authors claim in L278-279 that because of this, BAI is not appropriate but I don’t see why it would show opposite responses, something seems odd and should be explained properly. As stressed by Reviewer #2 and myself, the targeting of the trees is crucial. While the authors have improved the information on this issue, but it still not very clear. They said all trees were measured for dbh and height and then ‘randomly selected’ based on those. How that could be done? In a reply to another question in the cover letter, they suggest they targeted the trees that are close to the average dbh and height, but still remain random. That seems to contradict itself. Also, that means they specifically avoid the largest trees (canopy dominant?). In this regard, the authors acknowledged that targeting trees can create bias and cite Nehrbass-Ahles et al. work, however they go on to suggest that in plantations sample trees are representative of the growth trend (unbiased?) and thus useful to use while reducing the amount of work. Can you provide a reference that support this statement? I don’t see why planted forests would necessarily be free of the biases showed in many recent dendrochronological works. At the very least, these limitations and potential biases should be discussed in the main paper in a paragraph in the discussion. The rationale of the number of trees chosen should also be discussed. Very similarly, how the trees were selected for sapflow analyses is still not disclosed. In fact, there is very few discussion about the sapflow results currently, they are not mentioned in the abstract and barely in the discussion. They seem more an afterthought, please integrate this part with the rest better, as I think the sapflow measurements are the most novel part of this work. I am happy to see the authors will upload their data to open datasets, but there is not data availability statement, link, or repository information in the paper. This should be added and fully disclosed before acceptance, and a link to the data (dryad allows for private links while review is going on, I believe). Figure S1 is almost a copy of the main figure of Lloret et al. 2011, I would suggest them to include proper credits for it or remove it altogether. Figure 8 (former figure 7) is better this time but still not good. It needs to be specifically labelled as a schematic figure, which it is still not. I could see what the authors were going for with this, but I don’t think the figure helps more than showing the actual data. I would recommend either to re-do the graph to be actually schematic, reducing clutter and details and including mechanisms and general group (in fact they are still labelled as in the previous version with the old acronyms)m or otherwise, removing the figure. L333 claims ‘lower BAI than those of the LDT and BT groups (Fig. 4C), which was mainly associated with density-related competition’, I am not sure you can prove this conclusively without a proper test of all the other factors. It may be likely linked to it, but, as shown in Table 1, there are multiple factors that are different between these treatments, one of which is density. Where the plots were located is not shown now, just as photos in figure 1 and a general area. Can a simple topographical map with the locations of the plots be added to the supplementary? The main reason for this is to see how scattered or clustered these plots are and how much they differ in their microhabitat, which would help to understand the patterns observed and how much the differences could be influenced by other external factors.

Author Response

Please see the attachment, thanks.

Reviewer 2 Report

The revisions of the author's manuscript addressed most of my comments and recommendations.

Only my concerns regarding the age/size trend in the tree-ring width series were (in my opinion) not adequately addressed. The authors argued that:

'If RCS detrending was used, the difference of density effect also was removed.'

, which is only true for the absolute growth. For climate-growth-correlations only the relative growth (after detrending) is needed. Thus, using a flexible spline for detrending and then using the resulting tree-ring index for climate growth correlations can be used two answer which trees (high/low density) show a stronger relative reaction (percentage growth change in comparison to previous years). I am mainly raising this concern because Figure 3B looks exactly(!) like a typical size/age trend. In my first review I thus pointed out that this can lead to spurious correlations (e.g. with SPEI). I really do recommend that the authors additionally calculate climate growth correlations with 30-year spline detrended tree-ring width data and add this to the appendix. If this would show similar results as with the raw data (stronger correlations in high density plots), this would strongly corroborate the results.

Author Response

The revisions of the author's manuscript addressed most of my comments and recommendations.

Only my concerns regarding the age/size trend in the tree-ring width series were (in my opinion) not adequately addressed. The authors argued that:

'If RCS detrending was used, the difference of density effect also was removed.'

, which is only true for the absolute growth. For climate-growth-correlations only the relative growth (after detrending) is needed. Thus, using a flexible spline for detrending and then using the resulting tree-ring index for climate growth correlations can be used two answer which trees (high/low density) show a stronger relative reaction (percentage growth change in comparison to previous years). I am mainly raising this concern because Figure 3B looks exactly(!) like a typical size/age trend. In my first review I thus pointed out that this can lead to spurious correlations (e.g. with SPEI). I really do recommend that the authors additionally calculate climate growth correlations with 30-year spline detrended tree-ring width data and add this to the appendix. If this would show similar results as with the raw data (stronger correlations in high density plots), this would strongly corroborate the results.

Response: Thank you for your suggestion. We used the regional curve detrending (RCS) method to replace the negative exponential curve (NEL) method to redevelop a residual chronology. The residual chronology (RES) contains more high frequency information, and retains more growth climate information than the standard chronology (STD). RES chronology was used to assess the relationship of growth-climate in different densities plantation. We wouldn't have been improved our manuscript without your suggestion. Thank you very much again.

Round 2

Reviewer 1 Report

First, I would like to apologize to the authors and editors for the delay in my answer. I hope they understand that christmas in the western hemisphere is an important family and rest time and consequently work has to be many times delay until it finishes.

Having said that, I appreciate the authors efforts to include both of the reviewers’ comments, particularly regarding the inclusion of a standardized ring width section and its interpretation, which I think improves the work. I think the figures they modified or deleted were well chosen and they improved the paper. I am still personally not convinced at all about the authors lack of sampling strategy, I can agree with the authors that in this case, given the pictures they provided and the similarity of the trees with one another, it is likely not a major issue, but I would strongly recommend the authors to clearly disclose how plots and trees, meaning how the coordinates and size was selected, how each individual tree was decided to be collected or not and why the sampling sizes are what they are. Also, a data availability statement needs to be added to the text.

Other minor comments:

L30: this phrasing is wrong, maybe change for something like:

                “the HDT group showed significantly reduced whole-tree hydraulic capability, compared with the other two groups”

L32: change “indicate” to “suggest”

L33: is “completion index” correct? What does that mean? Please explain or correct

L54: “do not fully cover”… that is not correct English and I am not sure what you want to say, please correct and clarify.

L198: change “most likely” to “possibly”

L203: add “a” before “reduction”.

Author Response

Dear Reviewers,

Thank you for giving us the opportunity for revising the manuscript (ID: forests-677933). We thank the reviewers for the time and effort that they have put into reviewing the previous version of the manuscript. Their suggestions have enabled us to improve our work. We have studied the comments carefully and made corrections which we hope meet with approval. Revised portions are marked in red in the paper.

Appended to this letter is our point-by-point response to the comments raised by the reviewers. The comments are reproduced and our responses are given directly afterward in a red color.

We hope that all these corrections and revisions would be satisfactory. We would like also to thank you for allowing us to resubmit a revised copy of the manuscript.

We hope that the revised manuscript is accepted for publication.

Kind regards,

Shoujia Sun

Review comment

Having said that, I appreciate the authors efforts to include both of the reviewers’ comments, particularly regarding the inclusion of a standardized ring width section and its interpretation, which I think improves the work. I think the figures they modified or deleted were well chosen and they improved the paper. I am still personally not convinced at all about the authors lack of sampling strategy, I can agree with the authors that in this case, given the pictures they provided and the similarity of the trees with one another, it is likely not a major issue, but I would strongly recommend the authors to clearly disclose how plots and trees, meaning how the coordinates and size was selected, how each individual tree was decided to be collected or not and why the sampling sizes are what they are. Also, a data availability statement needs to be added to the text.

Response: Thanks for your suggestion. The sampled stands were located at Ertai Town Forest Farm (41.33 °N, 114.87 °E, elevation 1385 m). Three densities of Mongolian pine growing at the experimental location were assessed: high density stand (HDT) with 3m×3m intra-spacing, low density stand (LDT) with 4m×5m intra-spacing, and border trees (BT) growing on the edge of the stands or on the side of roads with neighboring natural grassland. Four plots (20 m × 20 m) were established in each of the three mature Mongolian pine plantation stands. For each plot, all trees were numbered, and then the diameter at breast height (DBH) and height were measured. Finally, total 12 sample plots in the three stands were surveyed. The Table1 showed the stand density, mean height, mean DBH and competition index. In each of the three stand densities, 6 trees in each plot and total 24 trees were randomly selected. All of the above were noted and marked in red in the manuscript.

We have added the data availability statement as “Data availability. The data used in this study are available as a supplementary file” at the end of the manuscript.( Current P14,L462)

We thank you very much for your time and effort that you have put into reviewing our manuscript. Your suggestions have enabled us to improve our work.

Other minor comments:

L30: this phrasing is wrong, maybe change for something like: “the HDT group showed significantly reduced whole-tree hydraulic capability, compared with the other two groups”

Response: Thanks for your good suggestions. It has been corrected in manuscript. ( Current P1,L30-31)

L32: change “indicate” to “suggest”

Response: it has been corrected. Thanks. ( Current P1,L32)

L33: is “completion index” correct? What does that mean? Please explain or correct

Response: I'm so sorry. It's my fault. It has been corrected as “competition”.  ( Current P1,L33) 

L54: “do not fully cover”… that is not correct English and I am not sure what you want to say, please correct and clarify.

Response: We correct it as “the specific responses of trees in stands of different densities to repeated drought events remains poorly understood”. ( Current P2,L53-54)

L198: change “most likely” to “possibly”

Response: it has been corrected as “possibly”. Thanks very much. ( Current P5,L200)

L203: add “a” before “reduction”.

Response: W have added “a” before “reduction”. Thanks again. ( Current P5,L204)

This manuscript is a resubmission of an earlier submission. The following is a list of the peer review reports and author responses from that submission.

Round 1

Reviewer 1 Report

In this paper entitled ‘Tree-ring analysis reveals density-dependent vulnerability to drought in planted Mongolian pines’, Sun and collaborators study the growth patterns of Mongolian pine (Pinus sylvestris mongolica) in the Inner Mongolian Plateau in Hebei, China. They focus on two main factors affecting Mongolian pine growth: competition (via stand density) and repeated drought events. To this end they divided the sampling plots in 3 categories: low density, high density, and border trees (24 plots in total, as stated in L133, although it’s unclear to me how many plots were assigned to each category). I understand that they sampled 24 trees in each of the categories but how these were selected or distributed among the plots is not described. They performed multiple analysis of growth trends and correlations to explore 3 main drought events they could identify based on climate data. They explored and compared the BAI and tree-ring trends of each of those groups, looked at their climate-growth correlations and calculated their resilience metrics using Lloret et al. 2011 approach. It is a nice adding that they also measured sap flow for some of their trees and found a clear reduction in sap flow in trees growing in high density, which supports their hypothesis.

The paper is correctly written and easy to understand. There are a handful of typos and wrong expressions (such as L16 should be corrected to ‘stand densities’; L20 ‘perbasal’ should be changed to ‘basal’, at least I’ve never seen ‘perbasal’ before myself; L94 2448°C I guess should be changed to ’24.48°C’; Fig. 3 grewth should be changed to ‘growth’; L320 Latin names should be in italics, and Populus nigra should be fully spelled out, as it is the first time that appears; L250 and 368 revise the use of ‘different’ vs ‘differences’, these sentences should read ‘were not different’ and ‘were not statistically different’, also in that one it should be changed to ‘all of the groups’, and some other ones here and there). These typos do not impede understanding the paper but would be great for the authors to revise the paper carefully in search of these easy mistakes.

On the content of the paper, the authors have done a good amount of work, on a relatively unstudied system, which is quite valuable. The result of a strong reduction in resilience after successive events in high density trees, which didn’t occur or was much lower in border and low-density trees, is a very interesting finding. Memory effects and variable interactions are important factors that are not frequently explored, and it is great to see them addressed in forest plantations, where they can have a key relevance. On the other hand, the manuscript misses, in its current form, multiple vital information to know that the analysis and interpretations are adequate. I would encourage the authors to re-write the manuscript, taking care to include enough information to make sure that the reader can properly interpret their findings. My main concerns with the paper are:

                - Lack of methodological clarity: Overall, many important methodological details are missing. How the trees were selected? how many trees were chosen per plot? how many plots per group? how these plots were established? How many of the trees were chosen for sap flow? Were the same as the ones used for dendrochronological analyses? How ‘border trees’ were identified? Meaning what were they bordering with? Natural grasslands? Human agriculture? They also talk about plots in multiple lines (L97, L133, etc.) but it is unclear what was measured at these plots. The methods section is currently long, and can be streamlined, but lack the crucial information for me to properly assess their work.

                - Very related to the previous point: How trees of each category were targeted can strongly affect the results, and it is not described at all. It is described in L102-104 that high density trees exhibited signs of decline. Were the trees chosen those with clear signs of decay? What was done to ensure they are representative of their categories? If they targeted the worst trees in HD category and the best in BT and LD categories, then the results seem almost unavoidable and may not represent the actual stand dynamics. Again, sampling strategies and design needs to be properly disclosed.

                - Please expand on why the design included ‘border trees’ and what was the expected result to come from them? It is also barely discussed why BT and LD differ so little between them, despite their strong difference in density, which is not in agreement with their initial hypothesis and needs to be properly discussed.

                - Figure 7 is unacceptable. This figure needs complete redoing. I don’t know if it’s supposed to be schematic (doesn’t say so) but it is plagued with errors, and it is unclear. Neither X nor Y axes have values or ticks, the values and the numbers associated with them do not correspond (0.15 is higher than 0.21?; 0.12 bigger than 0.17?; the distance between -0.01 and 0.01 seems bigger than that between 0.35 and 0.48; the stress events are not temporally homogeneous in reality, and so on). If this is a schematic figure, please re-do it in a clearer way and make clear that it is seen as a scheme (maybe values are not needed if it’s schematic?), and it still needs to make spatial sense.

                - I don’t know if it is a matter of the authors not submitting or a mistake in the editorial program, but I did not receive the supplementary material to assess. I would ask the authors or the editors to make sure to make this is available for the reviewers.

                - Finally, will this data be made public and available? As pointed out by many recent reviews, dendrochronological data from southeast Asia is deeply needed to improve our global ecological assessments. I would strongly encourage the authors to contribute to this effort by making their data freely available in some of the available platforms (dryad, ITRDB, DEN), and ‘Forests’ indeed requires that authors make their full datasets available whenever possible (see authors instructions in their website).

                Some other minor points that hopefully help further improving the manuscript:

This is a personal opinion, but acronyms are commonly overused and difficult to follow. This is certainly the case in this version of the manuscript. Maybe consider reducing them? Also, HD is an unfortunate acronym because in English is associated with ‘High Definition’, which makes a bit unsuitable to refer to ‘HD trees’, it would be a good idea to change these. However, as I said, this is a personal preference. L51-52. There has been an extensive work on drought response to trees and density, particularly from forestry, saying that ‘remain limited’ seems quite a stretch. Please refrain from using judgement terms such as ‘unsatisfactorily’. Replace with ‘not fully recover’ maybe? L65 and L69. Please change both these values to normal notation, which is easier to understand (and in this case even shorter! 3.83 x 104 is longer than just 38300). L81 ‘developed decline’ seems incorrect to me, ‘showed signs of decline’ maybe? Why you chose to measure 24 trees per category? seems a low number to characterize so many plots, particularly for the dendrochronological sampling. L144, please change the Columbia location reference for the cofecha program actual citation. How climate correlations were calculated? Did you use dendroclim? Treeclim package in R? Please specify. L189, please add the trend lines, because these trends are not obvious to me (particularly precipitation), are they significant? L206, ‘exhibited Λ-shaped profiles’ this seems a stretch given the variability, and LD and HD did not even show a hump, but just a decreasing trend according to your trend lines. Also, the emphasis on the analysis of the peak of the quadratic fitting seems questionable, seeing how variable they are. I would remove that part. L213, dates are in a different font size for no apparent reason, please correct. The interpretation of RRs is not very clear to me, could you explain what do actually mean for the tree to show lower RRs value? L228, the BAI of the HD showed higher growth than pre-drought after the first drought event? Surprising! How do you explain this? L275, what do you mean that BAI was a ‘biotic indicator’?

Reviewer 2 Report

Summary:

The authors present a well written manuscript and explore the effects of tree density and drought vulnerability in Mongolian pines. Both, drought vulnerability and tree density are highly important topics that attracted quite some attention in the past few years. Scope, sampling design and conclusions are plausible.

However, the study has some shortcomings in the data analysis, in particular the analysis of the climate sensitivity: Tree-ring width, as well as basal area increment are known to change with the size/age of a tree. These size/age trends can bias climate growth correlations. Because of that such data has to be detrended, for example with a flexible smoothing spline or regional curve standardization (RCS).

General notes:

- You are describing the climate in three sections: 2.1, 2.2 and 3.1. Since it is always just descriptive statistics (mean, extreme, trend…) I would recommend to merge it all into a single section in the 'methods' part

- Your chronologies start in 1988, but when were the trees planted?

- Figure 3-7: It is generally recommended not to use red and green in one graph since ~4% of people are red-green color blind

Comments by line:

- 20: Is 'perbasal' a typo? Just 'basal'?

- 93+: ' annual accumulated temperature 2448 °C ' - what temperatures did you accumulat? Hourly? Daily?, Weekly? Monthly? Anyway, I am not sure if this is a meaningful measure at all and you may leave it out. You probably used the same climate data as described in the next section but you should state here for what time period you have climate data: 1976-2018?

- 136: Table 1: What do the letters 'a', 'b', 'c' mean?

- 139: How did you select the trees? Largest DBH? Random? Mind the respective publications on sampling design: Nehrbass-Ahles et al. 2014, Xu et al. 2019

- 179: Why not also include previous year July, August and September? Some studies found the strongest correlations with previous year July climate. This is probably caused by accumulating NSCs for the next year

- 239-241: This sounds like you used raw tree-ring width date for the Pearson correlations. This will give you biased results because the long-term trend in the tree-ring series contains various noises. Most importantly, Figure 3b and 4b to me look like a clear tree-size/age trend!? So when tree-ring width declines over time due to size/age and SPEI declines over time you will get a spurious correlation between the two. You may mistake the size effect for an SPEI effect. For a simple climate sensitivity analysis you should at least detrend the tree-ring width series with a flexible smoothing spline. Personally I would even recommend to detrend the tree-ring data AND the climate data for this analysis. This way you will only correlate the 'high frequency signal'.

The same accounts for Table 2 and Figure 5a: please calculate climate growth correlations with detrended ring-width values. You could also use RCS detrending but I would recommend a spline.

- 243: measurement unit of the slope missing. Maybe 1.029mm/SPEI-unit?

- you write: ' which indicated that BAI was a biotic indicator related to the growth state and age of trees ' This is only partly true. Both, BAI and ring width, are the cumulative effect of many variables. This includes tree size/age, climate, nutrients, competition and so on. Differences in climate-growth correlations between BAI and ring width could be related to linear or non-linear (quadratic) effects.

- 323-324: ' A negative trend in BAI is a strong indicator of slowing tree growth[34]. ' - While this is true, you should consider that BAI peaks at a certain tree size/age, as explained above. Thus the question is if the growth decline is size/age related or drought related.

- 416: Figure 7 has no axes
